# *Dunaliella salina* as a Potential Biofactory for Antigens and Vehicle for Mucosal Application

Inkar Castellanos-Huerta [1], Gabriela Gómez-Verduzco [2], Guillermo Tellez-Isaias [3], Guadalupe Ayora-Talavera [4], Bernardo Bañuelos-Hernández [5,*], Víctor Manuel Petrone-García [6], Isidro Fernández-Siurob [7], Luis Alberto Garcia-Casillas [8] and Gilberto Velázquez-Juárez [8]

1  Programa de Maestria y Doctorado en Ciencias de la Produccion y de la Salud Animal, Facultad de Medicina Veterinaria y Zootecnia, Universidad Nacional Autonoma de Mexico, Ciudad Universitaria, Ciudad de Mexico C.P. 04510, Mexico
2  Departamento de Medicina y Zootecnia de Aves, Facultad de Medicina Veterinaria y Zootecnia, Universidad Nacional Autonoma de Mexico, Avenida Universidad 3000, Ciudad de Mexico C.P. 04510, Mexico
3  Department of Poultry Science, University of Arkansas, Fayetteville, AK 72701, USA
4  Centro de Investigaciones Regionales, Dr. Hideyo Noguchi, Universidad Autonoma de Yucatán (UADY), Mérida, Yucatán C.P. 97000, Mexico
5  Escuela de Veterinaria, Universidad De La Salle Bajío, Avenida Universidad 602, Lomas del Campestre, Leon C.P. 37150, Mexico
6  Departamento de Ciencias Pecuarias, Facultad de Estudios Superiores Cuautitlan UNAM, Cuautitlan C.P. 54714, Mexico
7  Viren SA de CV, Presidente Benito Juarez 110B, Jose Maria Arteaga, Queretaro, Queretaro C.P. 76135, Mexico
8  Departamento de Quimica, Centro Universitario de Ciencias Exactas e Ingenierias, Universidad de Guadalajara, Boulevard Marcelino Garcia Barragan #1421, Guadalajara, Jalisco C.P. 44430, Mexico
*  Correspondence: berbanher@gmail.com

**Abstract:** The demand for effective, low-cost vaccines increases research in next-generation biomanufacturing platforms and the study of new vaccine delivery systems (e.g., mucosal vaccines). Applied biotechnology in antigen production guides research toward developing genetic modification techniques in different biological models to achieve the expression of heterologous proteins. These studies are based on various transformation protocols, applied in prokaryotic systems such as *Escherichia coli* to eukaryotic models such as yeasts, insect cell cultures, animals, and plants, including a particular type of photosynthetic organisms: microalgae, demonstrating the feasibility of recombinant protein expression in these biological models. Microalgae are one of the recombinant protein expression models with the most significant potential and studies in the last decade. Unicellular photosynthetic organisms are widely diverse with biological and growth-specific characteristics. Some examples of the species with commercial interest are *Chlamydomonas*, *Botryococcus*, *Chlorella*, *Dunaliella*, *Haematococcus*, and *Spirulina*. The production of microalgae species at an industrial level through specialized equipment for this purpose allows for proposing microalgae as a basis for producing recombinant proteins at a commercial level. A specie with a particular interest in biotechnology application due to growth characteristics, composition, and protein production capacity is *D. salina*, which can be cultivated under industrial standards to obtain βcarotene of high interest to humans. *D saline* currently has advantages over other microalgae species, such as its growth in culture media with a high salt concentration which reduces the risk of contamination, rapid growth, generally considered safe (GRAS), recombinant protein biofactory, and a possible delivery vehicle for mucosal application. This review discusses the status of microalgae *D. salina* as a platform of expression of recombinant production for its potential mucosal application as a vaccine delivery system, taking an advance on the technology for its production and cultivation at an industrial level.

**Keywords:** *Dunaliella salina*; vaccines; recombinant protein; mucosal

## 1. Introduction

Nowadays, biotechnology applied to developing products in the health industry is highly diversified worldwide [1,2]; this interdisciplinary branch of biological sciences presents greater participation in the market of pharmaceuticals and vaccines each day [3,4]. Therefore, searching for new scientific developments with practical applications is a priority in the industry. The approaches of biotechnology applications on microalgae range from metabolic modification [5,6], to phytochemicals production (lipids, carbohydrates, terpenoids, phenolics, and alkaloids) [7], to the expression of recombinant proteins [8]. In this area, efforts are focused primarily on developing expression systems capable of achieving both the industrial aspects of production costs and the quality of the recombinant proteins expressed [8]. Through the development of protein expression systems, the study of biological models begins with prokaryotic models (*Streptomyces* spp., *Bacillus* spp., *Lactococcus lactis*, *Escherichia coli*, and *Corynebacterium glutamicum*) [9], in conjunction with different plasmid-based gene expression strategies, with advantages such as high level of protein expression, rapid cell division, low cost for the production of raw biomass.

Nevertheless, the prokaryotic system presents limitations such as forming insoluble inclusion bodies, purification process requirements due to the presence of endotoxins, and limited post-translational [10,11], which limit its use in the expression of high-quality proteins. Unlike prokaryotic systems, eukaryotic systems allow the possible design of complex proteins with post-translational characteristics [12,13], which in many cases compromise folding and function. Model eukaryotic expression platforms include yeast, animal, plant, mammalian/insect cells, and microalgae. Potvin et al., 2010 describe differences between these expression platforms, including (i) size of the heterologous gene, (ii) sensitivity to shear stress, (iii) recombinant product yield, (iv) production time, (v) cost of production, (vi) scale-up and storage cost. A complete comparison with several other production systems could be visualized in this review [14].

Due to its characteristics, the eukaryotic organism yeast is one of the expression systems widely used in the industry [15]. This organism presents advantages in its use, range of reproduction, culture in confinement (biological reactors), and an average cost of production; however, it has disadvantages such as post-translational modifications being significantly different from humans, and the high price of scale-up costs [14], require preliminary analysis to consider this system for the production of antigens. In the case of eukaryotic cell systems, including invertebrate and mammalian cell lines, advantages include the ability to achieve post-translational modifications resulting in high-quality products and efficient protein secretion to the medium, which facilitates purification and the availability of standard methods for genetic manipulation [16,17]. Nonetheless, its production cost, the high nutrient requirements, cell growth rate, and the risk of contamination by pathogens (viruses, bacteria, prions), in some cases, limit their use in vaccine production models [17]. A different approach to producing recombinant proteins is the generation of genetically modified organisms, e.g., plants and animals. The main advantage of using transgenic animals to produce recombinant proteins is the high yields of a high-quality product. Despite its benefits, the process of generating transgenic animals implies a long time between the genetic engineering phase and the start of production, a low rate of gene integration, and unpredictable behavior of the transgenes [18,19]. In particular, plants such as cereals, tobacco, legumes, fruits, and vegetables [19], present attractive advantages for their use in the production of recombinant proteins due to their production cost, the capacity of post-translational modifications, cultivation cost advantages, and low scale-up cost [8], but disadvantages such as production time, lacks regulatory approval [19], as well as the possible genetic contamination in populations of non-genetically modified plants [20], hinder its mass production. Plant cell culture in vitro represents an essential role as a new expression platform at the industrial level [21]. Plant cells, as they are not vectors of animal pathogens, viruses, prions, or bacteria, carry out complex post-translational modifications [22], as well as cultivation in closed systems, in addition to storing recombinant proteins at adequate levels stabilized by a simply freeze-drying process, show attractive advantages for its use

in mucosal vaccination. Nonetheless, this expression system is limited by the long period between the production of transgenic plants and recombinant proteins [21,23].

This outlook reflects the need for next-generation platforms to overcome some limitations in conventional systems. During the last two decades, microalgae have also emerged in this field as a potential new platform for the production of biopharmaceuticals [7,22] due to various advantages that are mentioned below.

## 2. Approaches for Mucosal Vaccine Delivery

The mucosal surface is a specialized tissue with the function of a selective barrier of the internal and external environment of organisms, capable of exchange of nutrients and oxygen and preventing the passage of foreign objects and pathogens. The protection of this particular tissue is mainly based on the participation of mucosa-associated lymphoid tissues (MALT), distributed along the mucosal surface [24], which are responsible for the production and secretion of a particular type of secretory immunoglobulin A (S-IgA). The predominant isotype in the local immune response in mucous membranes [25], S-IgA is an essential part of defenses at the local level, preventing the entry of pathogens [26]. Parenteral administration of antigens is not practical for their induction, so the mucosal immune system is considered separate from systemic immunity [27].

Interestingly, despite the advantages of mucosal vaccination such as non-invasiveness, mucosal solid immune response to prevent the entry of most infectious agents, ability to avoid the previous immune response by parenteral vaccination, local immune stimulation as systemic, as well as its easy application [25], mucosal vaccines approved for use are limited [26]. The design of a mucosal vaccine requires the correct selection of the following components: (1) an antigen capable of inducing an efficient immune response, (2) an adjuvant capable of stimulating the adaptive immune response, and (3) a suitable administration system. In the case of oral and nasal mucosal administration systems include viral vectors, virus-like particles, emulsions, immune-stimulating complexes, monophosphoryl lipid A, calcium phosphate nanoparticles, polymeric nanoparticles, liposomes, proteasomes, cholesterol-bearing pullulan nanoparticles, self-assembled peptides, nanogels, chitosan [25], plant tissue [27], and microalgae [28]. Microalgae models in recent years have reflected a development in the production of recombinant proteins, so its use as a vehicle for vaccine administration is interesting for research.

## 3. Microalgae as a Biofactory for Proteins

The denomination microalgae include all unicellular organisms with a photosynthetic capacity [29]. Therefore, this denomination comprises a broad polyphylogenic group, including species from cyanobacteria to eukaryotes [30]. The production of microalgae for human benefit is a practice known for over 2000 years [31]. Currently, the cultivation of microalgae has applications in the area of food, cosmetic, and pharmaceutical industries [32] because it constitutes a natural source of lipids, vitamins, pigments (zeaxanthin, lutein, astaxanthin, and phycocyanin), antimicrobials [33,34], and antioxidants [35]. Species with industrial interest include *Chlamydomonas*, *Botryococcus*, *Chlorella*, *Dunaliella*, *Haematococcus,* and *Spirulina* [29]. Another approach is protein production for human, and animal nutrition [35,36], due to high protein content [37,38]. In particular, the protein synthesis capacity of some microalgae species supports a possible use for the industrial production of recombinant proteins. The advantages of microalgae compared to other systems include a high growth rate, culture conditions in confined systems, availability of genetic engineering tools, absence of toxic compounds (generally recognized as safe GRAS classification), post-translational modifications, and high biosynthetic capacity in terms of biomass yield [14,33].

Regarding post-translational modifications, glycosylation is directly responsible for the immunogenicity of an antigen [39]. Therefore, a previous glycosylation patterns analysis of a system is necessary before selecting it for vaccine production. In general, protein expression systems possess species-specific N-glycans, with differences from human post-

translation modification including "hyper-mannosidosis" structures (excess of mannose residues assembled on yeast), absence of essential human residues ($\alpha$ (2,6)-sialic acid and $\alpha$(1,4)-fucose), undesired non-human residues (N-glycolylneuraminic acid (Neu5Gc) and galactose-$\alpha$ (1,3)-galactose ($\alpha$-Gal)) on CHO cells, glycans containing immunogenic residues ($\beta$(1,2)-xylose and core $\alpha$(1,3)-fucose) on plant cells. In the particular case of microalgae, two different glycosylation pathways are reported: GnT I enzyme-independent consists of 5 Man and 2 GlcNAc N-linked protein subjected to xylosyltransferases (XyT) and methyltransferases (MeT), leading to unique N-linked structures containing methylated mannoses linked to one or two xyloses with structures vary slightly among microalgae species, and GnT I enzyme-dependent transfers an N-acetylglucosamine residue to the 5 Mannose and 2 GlcNAc N-linked protein, subjected to $\alpha$-mannosidase II ($\alpha$-Man II) and fucosyltransferase (FuT), resulting in paucimannosidic (Man 3–4GlcNAc 2) fucosylated N-glycans. However, in these processes, microalgal species showed patterns more similar to humans [40].

In biopharmaceutical production, microalgae are a potential biofactory for antibodies, nanobodies, cytokines, antimicrobial peptides, vaccines, hormones, and enzymes [41,42]. A comprehensive list of proteins expressed in microalgae with industrial and biopharmaceutical applications in animals and humans has been reported in different species. Barolo et al., 2020 extensively describe proteins produced and the microalgae used in their essays [40].

Due to the industrial interest in microalgae cultivation, developing highly specialized systems focused on efficient and low-cost cultivation showed progress over the last decade [43–47]. Microalgae culture systems are classified into two types: open-type photobioreactors (raceways) and closed-type photobioreactors (PBR) [48]. These present systems differ in their design; however, their main concerns are the correct exposure to light, nutrients, temperature, and proper $O_2$ and $CO_2$ management [48]. Both designs are widely used. Therefore, producing biomass for recombinant proteins is a logical step. Some species considered possible expression platforms are *C. reinhardtii*, *C. vulgaris*, *C. ellipsoidea*, *D. salina*, *P. tricornutum*, and *N. oculata* [14,40,49].

Genetic engineering on microalgae proved potential for protein expression at the industrial level [18,40,50,51]. In addition, assays with microalgae species led to new protein expression platforms with distinct and innovative characteristics. Recently, *Dunaliella* sp. is one of the models proposed for study in protein expression [28,52–55]. The following subheadings describe the biological and industrial characteristics of *D. salina*.

## 4. *Dunaliella* sp. as a Production and Delivery Vehicle for Antigens

### 4.1. General Features

*Dunaliella* sp. is a unicellular, halophilic, biflagellate, naked green alga *Phylum Chlorophyta*, Class *Chlorophyceae*, order *Volvocales*, family *Polyblepharidaceae* with a total of 29 species, as well as several varieties and forms [56,57]. *D. salina* was first described in 1905 [58]. The genus *Dunaliella* sp., named in honor of Michel Felix Dunal [59] is the richest natural source of βcarotene, violaxanthin, neoxanthin, zeaxanthin, and lutein with the function of photoprotective to the high irradiance [60,61] and vitamins, antioxidants, polyunsaturated fatty acids, minerals, and enzymes [62]. Recently, the study of this species raised interest in its protein content, which ranges from 50 to 80% (dry weight), also for the content of essential amino acids, which is higher than recommended by the Food and Agriculture Organization of the United Nations (FAO) [36]. *Dunaliella* sp. presents different forms (spherical, pyriform, fusiform, ellipsoid), sizes from 5 to 25 μm in length and from 3 to 13 μm in width, also contain a single chloroplast, chlorophylls a and b, and organelles observed in green algae: membrane-bound nucleus, mitochondria, vacuoles, Golgi apparatus, and an eyespot and elastic plasma membrane covered by a mucus surface coat with the capacity of shrinks or swells according with the hypertonic and hypotonic conditions [62,63]. *D. salina*, similar to other microalgae, undergoes a complex life cycle, cellular divisions by lengthwise division in the motile state (vegetative cells), but also presents sexual reproduction (sexual zygote formation) [56].

Several species of *Dunaliella* sp. are observed in high salt concentrations, classifying them as halophilic organisms. However, some species thrive in freshwater [64,65] as well as over a wide pH range, from pH1 (*D. acidophila*) to pH11 (*D. salina*) [66]. The high capacity to adapt to different concentrations of salinity (3 to 31%) and temperature range (<0 °C to >38 °C) make *Dunaliella* sp. a unique and highly resistant eukaryotic organism [67]. Because of these characteristics, various species of *Dunaliella* sp. have been isolated in diverse ecosystems over the world [68]. Because of all high-value features, *Dunaliella* sp. can be considered a promisor recombinant expression system [49,63,69,70]; between these features are included: the capacity to grow in a wide range of salt concentrations which can prevent contamination of the culture [66], transcriptional modifications [42,50,71], and lacking a rigid cell wall, facilitating genetic transformation procedures as well as the extraction during downstream processing [63].

### 4.2. Production Aspects

*D. salina* culture media have wide ranges in salts and pH (6 to 23 % of NaCl, and pH 6 to 9) [56,72]. Optimal grown conditions are between 2 and 8% salt; a high salt concentration affects the growth rate in some cases. Under the best conditions for growth, the division rate can go from 0.5 to 1.22 divisions per 24 h [56]. Based on several studies, an average concentration of salt in the culture media of *D. salina* (12%) and *D. viridis* (6%) are the optimal salt concentration for growing [73,74]. However, other strains present different growth conditions [75]. In general, the culture conditions are a temperature of $25 \pm 2$ °C under the white fluorescent light of 52.84 μmol photons m$^{-2}$ s$^{-1}$ without aeration in stirring at 110 rpm/min in the orbital shaker [67,76]. The efforts focus on developing an efficient condition for growing under laboratory and industrial requirements [60,77–79]. Media for growth of *D. salina* suggested include: modified Johnson's medium, Erdschreiber's medium, Guillard's F/2 medium, modified ASP medium, and enriched seawater [76,80].

### 4.3. Culture Systems of D. salina

Mass culture of microalgae is reported in systems such as open ponds, circular ponds, raceway ponds, cascade ponds, large bags, tanks, heterotrophic fermenters, and several kinds of closed PBR [81,82]. In the case of *D. salina*, it can be grown under controlled conditions in selective media and biological contamination-free [83]. Currently, PBR implementation for *Dunaliella's* intensive culture is widely reported [63,84]. PBR has several advantages compared with other culture systems, such as higher yield, cleaner product, and concentration of secondary metabolites. In general, there are three types of PBR: flat plate bioreactors, tubular PBR, and ultrathin immobilized configurations [81,82,85]. The use of PBR for *Dunaliella* sp. culture has been focused on secondary metabolite production; however, their possible use as a PBR system for recombinant protein production is also feasible [86–89].

### 4.4. Genetic Engineering Tools Applied to D. salina

Among the genetic manipulation techniques reported for *Dunaliella* sp. include electroporation [90,91], particle bombardment [92], glass beads [93], lithium acetate/polyethylene glycol (PEG)-mediated method [55], and *Agrobacterium*-mediated method (agroinfiltration) [52]. In general, all techniques present a range of advantages and disadvantages for their use in microalgae [42]. Expression-efficacy depends on codon optimization, protease activity, protein toxicity, and transformation-associated genotypic modification [94]. In the case of *D. salina*, some of the technical approaches reported for nuclear transformation include LiAc/PEG-mediated method, glass bead method, and agroinfiltration protocol. In the case of chloroplast transformation, the most recommended method is particle bombardment. The possible use of other techniques, ultrasonic delivery [95], ultraviolet laser microbeam [96], and aerosol gene delivery [97], allows the opportunity to explore new approaches to achieve the best form of genetic manipulation in *Dunaliella* sp. These methods present a relatively low level of transformation and differences in their practicality and

repeatability; however, most of these are focused on the expression of reporters, selecting genes, therapeutic application, and production of viral proteins. Viral antigens, including hepatitis B surface antigen (HBsAg), yielding 3.11 ng/mg of total soluble protein by transforming electroporation protocol, white spot syndrome virus (WSSV) VP28, yielding 3.04 ng/mg of soluble protein by gene glass beads transformation [42], and hemagglutinin influenza virus yielding 255.5 µg/2 g wet weight by agroinfiltration protocol [52]. Despite the low expression levels [98], these assays are focused on determining the ability of this system to express viral proteins, so yields require other approaches.

One of the most promising systems for expressing recombinant proteins in *D. salina* is the agroinfiltration protocol mediated by *Agrobacterium tumefaciens* [52,99,100]. This protocol is based on the ability of *A. tumefaciens*, an indirect method, to transfer exogenous desoxyribonucleic acid (DNA) to plant cells through a bacterial conjugation system (Type IV secretion system (T4SS) and protein-DNA complexes) [101]. Plants are naturally affected by *A. tumefaciens*, including angiosperms and gymnosperms [102]. Briefly, *A. tumefaciens*, a bacterium present in the soil, moves towards the wound upon detecting phenolic compounds from a wounded plant, adheres, and begins to transform plant cells by inducing the transcription of virulence genes present in a plasmid called Tumor-inducer (Ti-DNA). Ti-DNA, together with the bacterial virulence proteins (VirD1, VirD2, VirE2), induces the transcription, processing of transfer DNA (T-DNA), and integration into the plant genome. Transferential DNA with *A. tumefaciens* requires the insertion of a gene of interest in T-DNA present in Ti-DNA for its insertion into the genome of the nucleus of the cells [102,103]. The random insertion observed in this method suggests a non-homologous recombination mechanism [104]. Since the first experiments for the elaboration of transgenic plants using *A. tumefaciens* in 1983 [102], significant advances in understanding the T-DNA insertion process, protocols, and experimentation in model plants, including in *D. salina* have been achieved.

### 4.4.1. Selection Markers and Reporter Genes

In the case of *D. salina*, selection markers, similar to antibiotics, require a different approach [92], because this microalga presents inherent resistance to a variety of antibiotics streptomycin, kanamycin, hygromycin (600 µg mL$^{-1}$) [90], spectinomycin (1200 mg L$^{-1}$) [100]. The use of chloramphenicol (60 µg mL$^{-1}$) [90], and zeocin 5 mg L$^{-1}$ [91] are feasible for the selection of transformed cells. Another selection gen for *D. salina* reported is herbicide phosphinothricin (PTT) (0.5 µg mL$^{-1}$) [105]. As reporter genes, the *gus* reporter gene [92], and the enhanced green fluorescent protein (EGFP), are applied [106].

### 4.4.2. Promoters and Enhancers for *D. salina*

The use of an efficient promoter is fundamental for the selection of any host. Among the principal exogenous promoters developed for *D. salina* are cauliflower mosaic virus 35, CaMV35S [55,91,92], Ubiquitin (Ubil), Ubil-Ω, CaMV35S-Ubil, CaMV35S-Ubil-Ω, endogenous promoters of actin gene [105], and glyceraldehyde-3-phosphate dehydrogenase [107], with high driving activity for gene expression. Inducible promoters include driving expression under a variety of sodium chloride concentrations (duplicated carbonic anhydrase 1 (DCA1)) [108], driving in the presence of nitrate, and inhibiting gene expression in the presence of ammonium ions (promoter *NR* gene) [109]. In the case of enhancers, a correct selection could prevent the gene silencing effect due to the position effect [110]. Enhancers reported on *D. salina* are the matrix attachment regions (MARs) [71] and 5′ leader sequence of tobacco mosaic virus RNA (Ω element) combined with promoters Ubil and CaMV35S-Ubil [90]. Studies suggest by MARs as a regulatory sequence increase expression levels [111,112], as well as stabilize their transcription processes [113,114] in transgenic offspring. Wang et al., 2005 demonstrated an increase in *CAT* gene expression 4.5-fold compared with other regulatory sequences through MARs in transformed *D. salina* [115,116]. Enhancers from other systems and genetic screening by selected UV-induced mutations with highly expressed nuclear transgenes open new possibilities [42,117]. In the case of

nuclear protein expression, random integration sites, RNA silencing, a compact chromatin structure, and non-conventional epigenetic effects [28,90,110], are possible factors affecting protein yield. Strategies to address these issues include surrounding insertion-site sequences analysis [118–120] and further study of regulatory sequences [111–114,121,122].

Currently, chloroplast transformation protocols in *D. salina* require a new approach due to potential observed in other organisms [42,123–126]; evaluation of regulatory sequences [42] and chloroplast transformation strategies [51,95,127–131] are possible improvement solutions. The publication of the complete sequence of the *D. salina* chloroplast genome (ptDNA) [132] encourages the development of more efficient transformation methods.

### 4.5. Advances in Dunaliella Transformation for Recombinant Biopharmaceutical Production

In general, expression in nucleus *D. salina* cells is focused mainly on reporter genes such as *β-glucuronidase* gene [50], enhanced green fluorescent protein [107], and selection markers such as phosphinothricin acetyltransferase under promoter DCA1 [110], chloramphenicol acetyltransferase [66], and zeocin resistance protein [105]. However, the expression of commercial value proteins is reduced [108,109], including immunogens [50]. Although several results [28,42,52,90], none of these proteins has led to the generation of products at the industrial level. According to findings, the *Dunaliella* system can be used in an approach for industrial applications, in particular in antigen production. The chloroplast is also an attractive expression protein system in microalgae due to advantages such as directed integration of genes via homologous recombination [133], high-level expression, organization of transgenes into operons, and no epigenetic interference [134,135], as previously reported [123,126]. Although there are few reports of expression in the chloroplast of *D. salina* [136], other systems such as *Chlamydomonas* [131], demonstrated that the use of chloroplast for the expression of recombinant proteins could be a proposal for proteins of commercial value. The purpose of new promoters and construction of expression vectors for *D. salina* chloroplast transformation is the following step [42] (Figure 1).

### 4.6. Immunological Aspects in Mucosal Vaccination with D. salina

Vaccination is one of the leading practices in medicine to control and prevent the vast majority of infectious-contagious diseases [137], based on the correct presentation of an antigen to the immune system. For this, it is necessary to determine the route of application, components of the formulation (adjuvant), type of immune responses, the dose required, and type of vaccine, either first group: (i) live attenuated vaccines (e.g., smallpox, yellow fever, measles, mumps, rubella, and chicken pox), or second group: (i) subunit vaccines (e.g., a vaccine against recombinant hepatitis B), (ii) toxoid vaccines (e.g., vaccines against diphtheria and tetanus), (iii) carbohydrate vaccines (e.g., vaccines against pneumococcus), and (iv) conjugate vaccines (e.g., vaccines against *Haemophilus influenzae* type B) [138].

Vaccination protocol and the immune system play a decisive role in correct immune response [139], particularly the immune system on the mucosal surface [140]. In general, the mucosal immune system presents highly specialized MALT, responsible for antigen presentation for the generation of an efficient mucosal immune response [24]. Due to the presence of these specialized tissues, mucosal administration of antigens demonstrated efficiency in wide pathologies [141], including influenza virus [142].

The expression of subunit vaccines in microalgae presents a convenient mucosal administration option with advantages including minimum processing before application [143] relatively low cost (<$1mg protein) in contrast to synthetic peptide antigen range between $35 and $95/mg peptide [143], algal cell wall appears sufficient to reduce antigen degradation by digestive system (bio-encapsulated) [98], subsequently broken down by digestive enzymes and commensal bacteria, and consequently, the recombinant proteins are released to be in contact with MALT [144,145]. Therefore, these drawbacks in consideration for oral vaccination are overcome by this expression model, as previously reported [28,52]. Considering that more than ten milligrams of an average subunit vaccine are required for oral administration (1000 times more than is necessary for an injected

route) [27], it is estimated that hundreds of grams of recombinant tissue are needed to stimulate an immune response. Nevertheless, increasing the concentration of the antigen by freeze-dried microalgae without losing antigenic capacity [27], antimicrobial activity [146], and immunomodulatory compounds naturally present in certain species of microalgae (*Dunaliella* sp.) exert synergistic effects with the vaccine formulation [147,148].

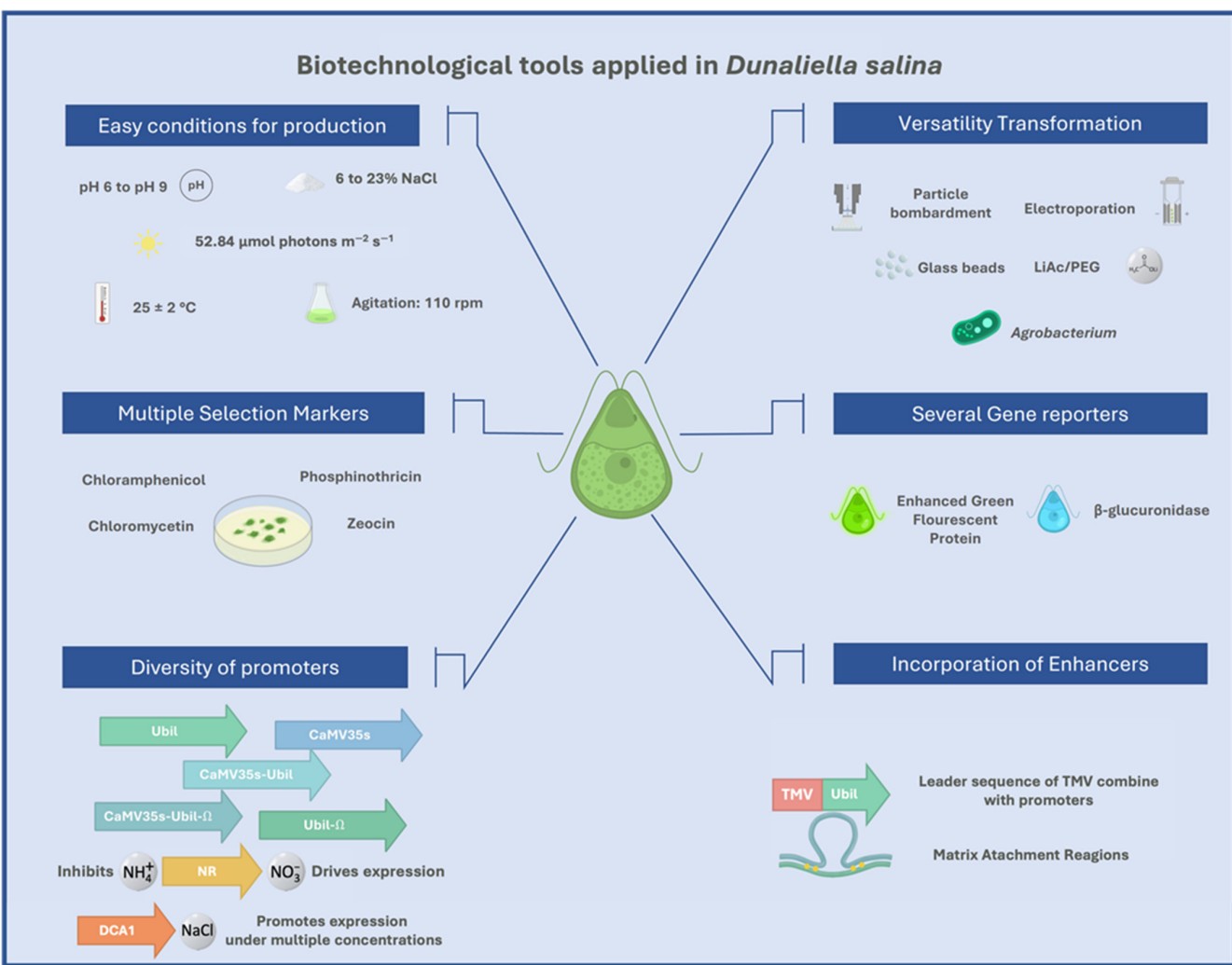

**Figure 1.** Biotechnological tools applied in *Dunaliella salina*.

## 5. Prospective View

Improvement in high-value recombinant protein expression systems encourages the research of several models with different advantages. Added to this, the cost of escalation and characteristics of post-translational modifications [14,149,150] are also aspects to consider.

In the case of *D. salina*, research showed this expression system as a practical solution for the production of different types of recombinant proteins and promises to be a production method based on the advantages of culture, transformation method, immunomodulatory compounds, glycosylation patterns, and natural encapsulation. The status of the investigation on the genetic manipulation of *D. salina* is in the early stages; however, the data suggest it can be a practical, tangible option for the vaccine industry. The mucosal administration of antigens (oral, ocular, or intranasal) presents requirements for a correct application, which not all recombinant protein expression systems can meet [14]. Hence, microalgae have advantages over other systems.

In addition to previously described advantages, microalgae are organisms capable of photosynthesizing by capturing environmental inorganic carbon ($CO_2$) [151]. As is known, microalgae are efficiently photosynthetic microorganisms [152]. Therefore, it is not only necessary to consider the advantages of a system such as microalgae without considering other social and environmental factors for developing a protein expression platform at an industrial level.

More research is needed on several details of the heterologous protein expression microalgae model, including regulatory sequences, codon-optimization [51,153], and efficient expression vectors [94,154,155]. Nevertheless, data available allows for considering *D. salina* as a protein expression system with potential for antigen production and its mucosal administration.

**Author Contributions:** Conceptualization, I.C.-H., G.G.-V., G.A.-T. and G.T.-I.; investigation, V.M.P.-G., G.V.-J. and L.A.G.-C.; resources, G.G.-V.; writing—original draft preparation, I.C.-H.; writing—review and editing, G.T.-I. and B.B.-H.; funding acquisition, I.F.-S. All authors have read and agreed to the published version of the manuscript.

**Funding:** This research was financed with private economic resources by the company VIREN SA DE CV.

**Institutional Review Board Statement:** Not applicable.

**Informed Consent Statement:** Not applicable.

**Data Availability Statement:** Not applicable.

**Conflicts of Interest:** Inkar Castellanos-Huerta is employed by Viren SA de CV. The remaining authors declare that the research was conducted in the absence of any commercial or financial relationships that could be construed as a potential conflict of interest.

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
