# Peer review of "Dunaliella salina as a Potential Biofactory for Antigens and Vehicle for Mucosal Application"

_processes, doi:10.3390/pr10091776_

Round 1
Reviewer 1 Report
This review, although with a title of Dunaliella salina as a potential biofactory for antigens and vehicle for mucosal application, failed to summarize the current state of research in using D. salina for antigen production and mucosal application. One third of the manuscript was spent on comparing different expression systems and the broad microalgae as biofactories. Particularly, mucosal was not mentioned until the line 378.
This review did not provide much of new information to readers compared to other reviews. For example, the authors claimed the table 1 and table 2 were updated from Potivin et al 2010 and Barolo et al 2020. However, they only added the columns mucosal delivery vehicle, safety and protein yield to table 1 and only hemagglutinin H5 to table 2 while the others are total the same.
The whole manuscript has many grammar errors which make it difficult to read. For example, the authors tend to write sentences in a format of “among+noun+verb” which is grammatically wrong. Line 300-304, the authors said the promote drives expression under different NaCl concentrations but then stated it induces gene expression in the presence of nitrate and ammonium ions, which is confusing.
The manuscript is wordy yet not much information has been given. For example, the authors spent a whole paragraph (from line 349 to 358) stating that chloroplast is advantageous as protein expression system. However, what kind of advantages or why it is advantageous was not given.
Author Response
Dear Reviewer, #1, thank you very much for the time you have spent reviewing our manuscript. Your comments are very valuable and helpful for revising our paper and guiding our research. We have studied those comments carefully and have made corrections, which we hope to meet with approval. The revised portion in the new version was included and is highlighted in yellow in the reviewed manuscript. The following is our point-by-point response to reviewers’ comments:
This review, although with a title of Dunaliella salina as a potential biofactory for antigens and vehicle for mucosal application, failed to summarize the current state of research in using D. salina for antigen production and mucosal application. One-third of the manuscript was spent on comparing different expression systems and the broad microalgae as biofactories. Particularly, mucosal was not mentioned until the line 378.
Your comments and observations are accepted and included in the text
This review did not provide much of new information to readers compared to other reviews. For example, the authors claimed the table 1 and table 2 were updated from Potivin et al 2010 and Barolo et al 2020. However, they only added the columns mucosal delivery vehicle, safety and protein yield to table 1 and only hemagglutinin H5 to table 2 while the others are total the same.
Your comments and observations are accepted and included in the text
The whole manuscript has many grammar errors which make it difficult to read. For example, the authors tend to write sentences in a format of “among+noun+verb” which is grammatically wrong. Line 300-304, the authors said the promote drives expression under different NaCl concentrations but then stated it induces gene expression in the presence of nitrate and ammonium ions, which is confusing.
Your comments and observations are accepted and included in the text
The manuscript is wordy yet not much information has been given. For example, the authors spent a whole paragraph (from line 349 to 358) stating that chloroplast is advantageous as protein expression system. However, what kind of advantages or why it is advantageous was not given.
Your comments and observations are accepted and included in the text

Reviewer 2 Report
This is a nice little review about the possible use of Dunaliella sp. as an expression host for the production of recombinant proteins, especially as recombinant vaccine antigens. However, the authors must improve the language; many words are misspelled, or a wrong word is chosen, and some sentences thus make no sense (although it is possible to guess what the authors wanted to say).
Furthermore, it would be great if the authors could elaborate more on posttranslational modifications, especially protein glycosylation, in Dunaliella sp. compared to other expression hosts. Proper glycosylation is often critical for good production yield and/or good antigenicity of the produced protein. What could be said about Dunaliella sp. or microalgae in general concerning this aspect?
Also, the whole idea of mucosal delivery by oral administration seems a bit challenging - the antigens enclosed within the microalgae cell that is (I presume) protected by a cell wall would be released only in the digestive tract (if at all) to be digested. What about pretreating the harvested microalgae biomass somehow to mildly disrupt the cell wall/cells and thus enable the antigen exposure right after the oral delivery? Is anything known/being tested in this regard?
Some parts of the review are pretty general, listing, e.g., all possible ways of gene insertion (transformation) into a cell. It would be great if, in the following parts of the text, the authors pinpoint in more detail which strategies were tested and successful in Dunaliella sp. or were not - it is only briefly mentioned with references, in a way that it seems easy to do - but I expect it is on the contrary quite tricky. So what I mean is to provide some more specific examples, perhaps taken from these references, such as what was the gene insertion rate/transfection efficiency, or in other places, what were the yields of the produced proteins, were those proteins biologically active, or towards which purpose were they used?
Author Response
Dear Reviewer, #2, thank you very much for the time you have spent reviewing our manuscript. Your comments are very valuable and helpful for revising our paper and guiding our research. We have studied those comments carefully and have made corrections, which we hope to meet with approval. The revised portion in the new version was included and is highlighted in yellow in the reviewed manuscript. The following is our point-by-point response to reviewers’ comments:
This is a nice little review about the possible use of Dunaliella sp. as an expression host for the production of recombinant proteins, especially as recombinant vaccine antigens. However, the authors must improve the language; many words are misspelled, or a wrong word is chosen, and some sentences thus make no sense (although it is possible to guess what the authors wanted to say).
Your comments and observations are accepted and included in the text
Furthermore, it would be great if the authors could elaborate more on posttranslational modifications, especially protein glycosylation, in Dunaliella sp. compared to other expression hosts. Proper glycosylation is often critical for good production yield and/or good antigenicity of the produced protein. What could be said about Dunaliella sp. or microalgae in general concerning this aspect?
Your comments and observations are accepted and included in the text
Also, the whole idea of mucosal delivery by oral administration seems a bit challenging - the antigens enclosed within the microalgae cell that is (I presume) protected by a cell wall would be released only in the digestive tract (if at all) to be digested. What about pretreating the harvested microalgae biomass somehow to mildly disrupt the cell wall/cells and thus enable the antigen exposure right after the oral delivery? Is anything known/being tested in this regard?
Your comments and observations are accepted and included in the text
Some parts of the review are pretty general, listing, e.g., all possible ways of gene insertion (transformation) into a cell. It would be great if, in the following parts of the text, the authors pinpoint in more detail which strategies were tested and successful in Dunaliella sp. or were not - it is only briefly mentioned with references, in a way that it seems easy to do - but I expect it is on the contrary quite tricky. So what I mean is to provide some more specific examples, perhaps taken from these references, such as what was the gene insertion rate/transfection efficiency, or in other places, what were the yields of the produced proteins, were those proteins biologically active, or towards which purpose were they used?
Your comments and observations are accepted and included in the text

Reviewer 3 Report
The topic is interesting. However, regarding to the title of the manuscript, the manuscript has not been focused on the antigens and mucosal applications well. In addition, it must be required to compare the potential of D. salina for these applications to other sources.
Author Response
Dear Reviewer, #3, thank you very much for the time you have spent reviewing our manuscript. Your comments are very valuable and helpful for revising our paper and guiding our research. We have studied those comments carefully and have made corrections, which we hope to meet with approval. The revised portion in the new version was included and is highlighted in yellow in the reviewed manuscript. The following is our point-by-point response to reviewers’ comments:
The topic is interesting. However, regarding to the title of the manuscript, the manuscript has not been focused on the antigens and mucosal applications well. In addition, it must be required to compare the potential of D. salina for these applications to other sources.
Your comments and observations are accepted and included in the text

Round 2
Reviewer 1 Report
The revised review did show certain improvements. However, there are still some problems listed below.
1. The sentence from line 131-136 has no subject or verb which make it hard to understand.
2. The paragraph from line 156-167 focused on post-translational modifications. Please elaborate on the PTMs in microalgae rather than other organisms.
3. The section 4.4.1 is about the selection markers. It would be valuable to the readers to add the concnetrations for selections in D. salina.
4. In the paragraph from line 373-394, the authors first mentioned the expression of subunit vaccines in plant tissues presents some issues whereas that in microalgae showed advantages. I was expecting the authors to elaborate its advantages. However, the authors instead discussed about the plant tissue for mucosal vaccines, which is irrelevant to the focus of this review ---- D. salina or microalgae.
5. There are still many grammar errors throughout the reviews (subject or verb missing). I will leave this part to the editor.
Author Response
Dear Reviewer # 1, thank you very much for the time you have spent reviewing our manuscript. Your comments are very valuable and helpful for revising our paper and guiding our research. We have studied those comments carefully and have made corrections, which we hope to meet with approval. The revised portion in the new version was included and is highlighted in yellow in the reviewed manuscript. The following is our point-by-point response to reviewers’ comments:
The revised review did show certain improvements. However, there are still some problems listed below.
- The sentence from line 131-136 has no subject or verb, which make it hard to understand.
We appreciate your comments and suggestions, which are accepted and included.
- The paragraph from lines 156-167 focused on post-translational modifications. Please elaborate on the PTMs in microalgae rather than other organisms.
We appreciate your comments and suggestions, which are accepted and included.
- Section 4.4.1 is about the selection markers. It would be valuable to the readers to add the concentrations for selections in D. salina.
We appreciate your comments and suggestions, which are accepted and included.
- In the paragraph from line 373-394, the authors first mentioned the expression of subunit vaccines in plant tissues presents some issues whereas that in microalgae showed advantages. I was expecting the authors to elaborate its advantages. However, the authors instead discussed about the plant tissue for mucosal vaccines, which is irrelevant to the focus of this review ---- D. salina or microalgae.
We appreciate your comments and suggestions, which are accepted and included.
- There are still many grammar errors throughout the reviews (subject or verb missing). I will leave this part to the editor.
We appreciate your comments and suggestions, which are accepted and included.

Reviewer 2 Report
The revised version of the manuscript is undoubtedly much better in terms of proper use of the English language and readability. However, my other comments and suggestions were only partially addressed. The authors provided three specific examples of recombinant proteins produced in Dunaliella and their yields. However, I do not fully understand what they mean by, e.g., "3.11 ng/mg of soluble protein" - 3.11 ng of HBsAg in 1 mg of total soluble protein in Dunaliella cell? Well, the point is that this tells nothing to the reader if a proper comparison is not given as well, i.e., what is the production yield for HBsAg in other hosts/expression systems? Ultimately, is Dunaliella indeed a promising host or not, comparing the yields? The glycosylation of proteins produced in Dunaliella is not discussed, as well as my concerns about oral delivery, or only in a very general way. Overall, reading this review, I learned that it is, in principle, possible to express recombinant proteins in Dunaliella, but why on earth should it be better than other systems? Maybe the costs could be lower because it is an autotrophic organism - but this is not supported by data in the review, e.g., comparison of costs, yields, other benefits... Yes, we can eat algae, that is for sure - but do we digest it all? Can we digest its cell wall? What is the envisioned way of antigen delivery? Any specific examples there? What I am trying to say is that instead of only citing other works, it would be great if the reader of this review could learn the answers directly from the review, without the need to read the other works to find them out on their own.
Author Response
Dear Reviewer # 2, thank you very much for the time you have spent reviewing our manuscript. Your comments are very valuable and helpful for revising our paper and guiding our research. We have studied those comments carefully and have made corrections, which we hope to meet with approval. The revised portion in the new version was included and is highlighted in yellow in the reviewed manuscript. The following is our point-by-point response to reviewers’ comments:
The revised version of the manuscript is undoubtedly much better in terms of proper use of the English language and readability.
However, my other comments and suggestions were only partially addressed.
The authors provided three specific examples of recombinant proteins produced in Dunaliella and their yields. However, I do not fully understand what they mean by, e.g., "3.11 ng/mg of soluble protein" - 3.11 ng of HBsAg in 1 mg of total soluble protein in Dunaliella cell? Well, the point is that this tells nothing to the reader if a proper comparison is not given as well, i.e., what is the production yield for HBsAg in other hosts/expression systems? Ultimately, is Dunaliella indeed a promising host or not, comparing the yields. The glycosylation of proteins produced in Dunaliella is not discussed, as well as my concerns about oral delivery, or only in a very general way. Overall, reading this review, I learned that it is, in principle, possible to express recombinant proteins in Dunaliella, but why on earth should it be better than other systems?
Maybe the costs could be lower because it is an autotrophic organism - but this is not supported by data in the review, e.g., comparison of costs, yields, other benefits... Yes, we can eat algae, that is for sure - but do we digest it all? Can we digest its cell wall? What is the envisioned way of antigen delivery? Any specific examples there? What I am trying to say is that instead of only citing other works, it would be great if the reader of this review could learn the answers directly from the review, without the need to read the other works to find them out on their own.
We appreciate your comments and suggestions, which are accepted and included in the text.

Reviewer 3 Report
I recommend for accepting this manuscript.
Author Response
Dear Reviewer # 3, thank you very much for the time you have spent reviewing our manuscript. Your comments are very valuable and helpful for revising our paper and guiding our research. We have studied those comments carefully and have made corrections, which we hope to meet with approval. The revised portion in the new version was included and is highlighted in yellow in the reviewed manuscript. The following is our point-by-point response to reviewers’ comments:
We appreciate your comments and suggestions, which are accepted and included in the text.